# Multiple cardiovascular risk factor care in 55 low- and middle-income countries: A cross-sectional analysis of nationally-representative, individual-level data from 280,783 adults

**Alpha Oumar Diallo**[1], **Maja E. Marcus**[2], **David Flood**[3], **Michaela Theilmann**[4], **Nicholas E. Rahim**[5], **Alan Kinlaw**[6,7], **Nora Franceschini**[1], **Til Stürmer**[1], **Dessie V. Tien**[5], **Mohsen Abbasi-Kangevari**[8], **Kokou Agoudavi**[9], **Glennis Andall-Brereton**[10], **Krishna Aryal**[11], **Silver Bahendeka**[12], **Brice Bicaba**[13], **Pascal Bovet**[14,15], **Maria Dorobantu**[16], **Farshad Farzadfar**[8], **Seyyed-Hadi Ghamari**[8], **Gladwell Gathecha**[17], **David Guwatudde**[18], **Mongal Gurung**[19], **Corine Houehanou**[20], **Dismand Houinato**[20], **Nahla Hwalla**[21], **Jutta Jorgensen**[22], **Gibson Kagaruki**[23], **Khem Karki**[24], **Joao Martins**[25], **Mary Mayige**[23], **Roy Wong McClure**[26], **Sahar Saeedi Moghaddam**[27], **Omar Mwalim**[28], **Kibachio Joseph Mwangi**[17], **Bolormaa Norov**[29], **Sarah Quesnel-Crooks**[10], **Abla Sibai**[30], **Lela Sturua**[31], **Lindiwe Tsabedze**[32], **Chea Wesseh**[33], **Pascal Geldsetzer**[4,34], **Rifat Atun**[35,36], **Sebastian Vollmer**[2], **Till Bärnighausen**[4,35,37], **Justine Davies**[38,39,40], **Mohammed K. Ali**[41], **Jacqueline A. Seiglie**[42], **Emily W. Gower**[1,43]☯*, **Jennifer Manne-Goehler**[2,44]☯

1 Department of Epidemiology, Gillings School of Global Public Health, University of North Carolina at Chapel Hill, Chapel Hill, North Carolina, United States of America, 2 Department of Economics and Centre for Modern Indian Studies, University of Goettingen, Göttingen, Germany, 3 University of Michigan, Ann Arbor, Michigan, United States of America, 4 Faculty of Medicine and University Hospital, Heidelberg Institute of Global Health, Heidelberg University, Heidelberg, Germany, 5 Medical Practice Evaluation Center, Massachusetts General Hospital, Harvard Medical School, Boston, MA, United States of America, 6 Division of Pharmaceutical Outcomes and Policy, Eshelman School of Pharmacy, University of North Carolina School of Pharmacy at Chapel Hill, Chapel Hill, North Carolina, United States of America, 7 Cecil G. Sheps Center for Health Services Research, University of North Carolina at Chapel Hill, Chapel Hill, North Carolina, United States of America, 8 Non-Communicable Diseases Research Center, Endocrinology and Metabolism Population Sciences Institute, Tehran University of Medical Sciences, Tehran, Iran, 9 Togo Ministry of Health, Lome, Togo, 10 Caribbean Public Health Agency, Port of Spain, Trinidad and Tobago, 11 Nepal Health Sector Programme 3, Monitoring Evaluation and Operational Research Project, Abt Associates, Kathmandu, Nepal, 12 Saint Francis Hospital Nsambya, Kampala, Uganda, 13 Institut Africain de Santé Publique, Ouagadougou, Burkina Faso, 14 Ministry of Health, Victoria, Seychelles, 15 University Center for Primary Care and Public Health (Unisanté), Lausanne, Switzerland, 16 Department of Cardiology, Emergency Hospital of Bucharest, Bucharest, Romania, 17 Division of Non-Communicable Diseases, Ministry of Health, Nairobi, Kenya, 18 Department of Epidemiology and Biostatistics, School of Public Health, Makerere University, Kampala, Uganda, 19 Health Research and Epidemiology Unit, Ministry of Health, Thimphu, Bhutan, 20 Laboratory of Epidemiology of Chronic and Neurological Diseases, Faculty of Health Sciences, University of Abomey-Calavi, Cotonou, Benin, 21 Faculty of Agricultural and Food Sciences, American University of Beirut, Beirut, Lebanon, 22 Dept of Public Health and Epidemiology, Institute of Global Health, Copenhagen University, Copenhagen, Denmark, 23 National Institute for Medical Research, Dar es Salaam, Tanzania, 24 Department of Community Medicine and Public Health, Institute of Medicine, Tribhuvan University, Kathmandu, Nepal, 25 Faculty of Medicine and Health Sciences, Universidade Nacional Timor Lorosa'e, Dili, Timor-Leste, 26 Office of Epidemiology and Surveillance, Costa Rican Social Security Fund, San José, Costa Rica, 27 Endocrinology and Metabolism Research Center, Endocrinology and Metabolism Clinical Sciences Institute, Tehran University of Medical Sciences, Tehran, Iran, 28 Ministry of Health, Zanzibar City, Tanzania, 29 Nutrition Department, National Center for Public Health, Ulaanbaatar, Mongolia, 30 Department of Epidemiology and Population Health, Faculty of Health Sciences, American University of Beirut, Beirut, Lebanon, 31 Non-Communicable Disease Department, National Center for Disease Control and Public Health, Tbilisi, Georgia, 32 Ministry of Health, Mbabane, Eswatini, 33 Ministry of Health, Monrovia, Liberia, 34 Division of Primary Care and Population Health, Stanford University, Stanford, California, United States of America, 35 Department of Global Health and Population, Harvard T.H. Chan School of Public Health, Harvard University, Boston, Massachusetts, United States of America, 36 Department of Global Health and Social Medicine, Harvard Medical School, Harvard University, Boston,



**Data Availability Statement:** Data included in this study are publicly available for 35 of the 56 countries. Microdata can be downloaded (upon

free registration) for Bangladesh 2018, India 2015-2016, and Namibia 2013 at the following website: https://dhsprogram.com/. The country surveys included in this analysis that are publicly available through the STEPS Microdata repository (https://extranet.who.int/ncdsmicrodata/index.php/catalog/STEPS) are: Algeria 2016, Azerbaijan 2017, Belarus 2016, Benin 2015, Botswana 2014, Cambodia 2010, Eritrea 2010, Iraq 2015, Kiribati 2015, Kyrgyzstan 2013, Laos 2013, Lebanon 2017, Lesotho 2012, Marshall Islands 2017, Moldova 2013, Mongolia 2013, Morocco 2017, Myanmar 2014, Rwanda 2012, Samoa 2013, Sao Tome and Principe 2009, Solomon Islands 2015, Sri Lanka 2014, Sudan 2016, Tajikistan 2016, Tuvalu 2015, Vietnam 2015, Zambia 2017. The following five surveys can be accessed at their specific websites: Chile 2009-2010: http://epi.minsal.cl/encuesta-ens-anteriores/; China 2009: http://www.cpc.unc.edu/projects/china/data; Ecuador 2018: http://www.ecuadorencifras.gob.ec/salud-salud-reproductiva-y-nutricion/; Indonesia 2014: https://www.rand.org/labor/FLS/IFLS.html; Mexico 2009-2012: www.ennvih-mxfls.org/english/index.html. For the remaining countries, please contact ghp@hsph.harvard.edu. For Guyana and St. Vincent and the Grenadines, which are member countries of the Caribbean Public Health Agency (CARPHA): Data were originally shared through a Data Use Agreement signed with the Executive Director of CARPHA and the agreement of The Ministries of Health of St. Vincent and the Grenadines and Guyana. The Chief Medical Officer of the St. Vincent and the Grenadine's Ministry of Health (Dr. Simone Keizer-Beache) can be contacted, if necessary.

**Funding:** Harvard T H Chan School of Public Health McLennan Fund: Dean's Challenge Grant Program and the EU's Research and Innovation programme Horizon 2020. There are no grant numbers. The funders had no role in study design, data collection and analysis, decision to publish, or preparation of the manuscript.

**Competing interests:** The authors have declared that no competing interests exist.

Massachusetts, United States of America, **37** Africa Health Research Institute, Somkhele, South Africa, **38** MRC/Wits Rural Public Health and Health Transitions Research Unit, School of Public Health, University of Witwatersrand, Johannesburg, South Africa, **39** Institute of Applied Health Research, University of Birmingham, Birmingham, United Kingdom, **40** Centre for Global Surgery, Department of Global Health, Stellenbosch University, Cape Town, South Africa, **41** Hubert Department of Global Health, Rollins School of Public Health, Emory University, Atlanta, Georgia, United States of America, **42** Diabetes Unit, Massachusetts General Hospital, Boston, MA, United States of America, **43** Department of Ophthalmology, School of Medicine, University of North Carolina at Chapel Hill, Chapel Hill, North Carolina, United States of America, **44** Division of Infectious Diseases, Brigham and Women's Hospital, Harvard Medical School, Boston, Massachusetts, United States of America

☯ These authors contributed equally to this work.
* egower@unc.edu

## Abstract

The prevalence of multiple age-related cardiovascular disease (CVD) risk factors is high among individuals living in low- and middle-income countries. We described receipt of healthcare services for and management of hypertension and diabetes among individuals living with these conditions using individual-level data from 55 nationally representative population-based surveys (2009–2019) with measured blood pressure (BP) and diabetes biomarker. We restricted our analysis to non-pregnant individuals aged 40–69 years and defined three mutually exclusive groups (i.e., hypertension only, diabetes only, and both hypertension-diabetes) to compare individuals living with concurrent hypertension and diabetes to individuals with each condition separately. We included 90,086 individuals who lived with hypertension only, 11,975 with diabetes only, and 16,228 with hypertension-diabetes. We estimated the percentage of individuals who were aware of their diagnosis, used pharmacological therapy, or achieved appropriate hypertension and diabetes management. A greater percentage of individuals with hypertension-diabetes were fully diagnosed (64.1% [95% CI: 61.8–66.4]) than those with hypertension only (47.4% [45.3–49.6]) or diabetes only (46.7% [44.1–49.2]). Among the hypertension-diabetes group, pharmacological treatment was higher for individual conditions (38.3% [95% CI: 34.8–41.8] using antihypertensive and 42.3% [95% CI: 39.4–45.2] using glucose-lowering medications) than for both conditions jointly (24.6% [95% CI: 22.1–27.2]).The percentage of individuals achieving appropriate management was highest in the hypertension group (17.6% [16.4–18.8]), followed by diabetes (13.3% [10.7–15.8]) and hypertension-diabetes (6.6% [5.4–7.8]) groups. Although health systems in LMICs are reaching a larger share of individuals living with both hypertension and diabetes than those living with just one of these conditions, only seven percent achieved both BP and blood glucose treatment targets. Implementation of cost-effective population-level interventions that shift clinical care paradigm from disease-specific to comprehensive CVD care are urgently needed for all three groups, especially for those with multiple CVD risk factors.

## Introduction

Cardiovascular diseases (CVD), including ischemic heart disease and stroke, are the leading causes of morbidity and mortality worldwide. Low- and middle-income countries (LMICs)

have higher burdens of CVD [1–3], largely due to growing populations and increasing life expectancy putting more people at risk of developing multiple risk factors [4]. Coexisting CVD risk factors (e.g., hypertension, diabetes, hyperlipidemia) increase the complexity of symptom management needed to prevent further deterioration of quality of life and health [5, 6], contributing to increased morbidity and mortality [7, 8]. The rising prevalence of individuals with multiple CVD risk factors creates a growing urgency for health systems in LMICs to provide comprehensive CVD care [9–11], especially to those at highest risk, to prevent cardiovascular-related morbidity and mortality [12, 13].

The 75th World Health Assembly (2022) ratified global coverage targets for diabetes to prevent diabetes-related complications as part of high-level efforts to reach global sustainable development goals (SDG) target 3.4, which calls for a reduction in premature mortality from non-communicable diseases (NCDs) by a third by 2030 relative to 2015 levels, through diabetes prevention and management [14]. Global coverage targets were established for diabetes because it is a major obstacle to achieving SDG target 3.4 given that it is strongly associated with other CVD risk factors like hypertension and hyperlipidemia and is a leading cause of mortality [15–17]. The global diabetes coverage targets are that by 2030, 80% of people living with diabetes are clinically diagnosed; and among them, 80% have good control of glycemia (hemoglobin $A_{1c}$ <8%), and 80% have good control of blood pressure (<140/90 mm Hg), and 60% of those with diabetes aged 40 years or older are taking a statin; 100% of those with type 1 diabetes have access to affordable insulin and blood glucose self-monitoring [14]. These targets highlight the increasing recognition that CVD risk factors including hypertension, diabetes, and hyperlipidemia should be managed concurrently. Additionally, the Assembly agreed on recommendations to increase health systems' capacity to deliver cost-effective population-wide interventions and monitor progress towards these targets.

Recent analyses of health system performance in LMICs have largely focused on a single CVD risk factor [18–24], and found that individuals' awareness of their condition, medication use, and achievement of recommended treatment goals (i.e. improved management of their condition) are suboptimal—approximately 10% of people with hypertension and 23% of those with diabetes achieved treatment goals, respectively [18, 19, 25]. Few studies have examined the receipt of healthcare services (e.g., diagnosis and treatment) and the management of conditions for CVD risk reduction of among those with multiple CVD risk factors in LMICs [26, 27].

We examined whether individuals with both hypertension and diabetes were more likely to receive healthcare services and appropriately manage their conditions than those with just one of these conditions using individual-level data across multiple countries to provide a benchmark for the global coverage targets for diabetes by 2030 and the promotion of comprehensive CVD care. We also assessed appropriate management of hyperlipidemia with lipid-lowering medications (statins) as part of diabetes management according to World Health Organization (WHO) guidance [28, 29].

## Methods

### Study design and participants

We analyzed pooled, cross-sectional survey data identified through the Global Health and Population Project on Access to Care for Cardiometabolic Disease (HPACC)'s search methodology [30] that: (1) were conducted in 2009 or later in an LMIC as classified by the World Bank in the survey year; (2) were nationally representative; (3) had individual-level data available; (4) contained physiological measures of blood pressure (BP) and either blood glucose

(BG) or hemoglobin A1c (HbA1c); and (6) had a response rate of ≥50%. This resulted in 55 eligible surveys conducted during 2009–2019, most of which (n = 47) used the WHO recommended Stepwise Approach to Non-Communicable Disease (NCD) Risk Factor Surveillance (STEPS) instruments for population monitoring of NCD targets.

Our sample consisted of non-pregnant individuals 40–69 years to align with the STEPS surveys inclusion criteria, most of which set an age limit of 69, and CVD management recommendation by the WHO Package of Essential Noncommunicable (PEN) Disease Interventions for primary health care in low-resource settings [28]. The PEN outlines instructions for the assessment, diagnosis, treatment, and management of diabetes and hypertension, aligning with leading clinical guidelines and prevention recommendations [28, 31, 32].

## Outcomes and procedures

We defined three mutually-exclusive study groups–namely hypertension only, diabetes only, and both hypertension and diabetes (henceforth referred to as hypertension-diabetes)–to compare individuals with concurrent hypertension and diabetes to individuals with each of these conditions separately. *Hypertension* was defined as either: systolic BP (SBP) ≥140 mmHg or diastolic BP (DBP) ≥90 mmHg as per survey biomarker measurement, self-reported antihypertensive medication use, or self-reported diagnosis by a clinician [28]. *Diabetes* was defined as either: fasting plasma glucose (FPG) ≥7.0 mmol/L (126 mg/dL), random plasma glucose ≥11.1 mmol/L (200 mg/dL) or HbA1c ≥6.5% as per survey biomarker measurement, self-reported glucose-lowering medication use, or self-reported diagnosis by a clinician [28]. **S1–S4 Tables** provides further details the hypertension and diabetes definitions as well as the BP and diabetes biomarker measurements.

For each of these study groups, we derived four main outcomes based on the PEN recommendation and the global diabetes coverage targets: awareness of diagnosis, receipt of lifestyle counseling, receipt of pharmacological therapy, and achievement of appropriate hypertension and diabetes management (see **Table 1** and **S5 Table**).

*Awareness of diagnosis* was defined by respondents reporting having been told by a doctor or healthcare worker that they have elevated BP or BG.

The *lifestyle counseling* outcome was defined as respondents reporting to being advised to (1) start or increase physical activity, (2) reduce salt intake, or (3) maintain a healthy body weight or lose weight during any visit to a doctor or healthcare worker in the past 12 months. Physical activity and weight loss were assessed for all study groups and salt reduction was examined among those with hypertension only or hypertension-diabetes.

The *pharmacological therapy* outcome was defined as self-reported use of antihypertensive, glucose-lowering, or cholesterol-lowering (statin) medications in the past two weeks, individually or in combination. We described statin use among the diabetes only and hypertension-diabetes groups, since statins are recommended in individuals aged ≥40 years with diabetes, regardless of lipid values, for primary prevention of CVD [28].

To assess *appropriate hypertension and diabetes management*, we applied the WHO PEN recommended treatment goals to the BP and diabetes biomarker measurements [33]. The BP treatment goal was set at SBP <140 mmHg and DBP <90 mmHg. The diabetes treatment goal was based on HbA1c <7.0% or FPG <7.0 mmol/L (<126 mg/dl) if HbA1c was not available. We used the PEN HbA1c definition <7.0% rather than the global diabetes coverage target of <8.0% because we included adults aged 40–69 years; a higher target is often chosen as a blanket metric when older adults are part of the denominator. Among the diabetes only and hypertension-diabetes groups, we defined an additional control indicator that combined the diabetes treatment goal defined above and the self-reported use of statins.

**Table 1. Definition of outcomes and the study groups among whom they were recommended.**

| Outcome | Study group[b] | | |
|---|---|---|---|
| Indicator[a] | Hypertension Only | Diabetes Only | Hypertension-Diabetes |
| **Diagnosis** | | | |
| Awareness of condition | X | X | X |
| *Individuals self-reporting to have ever been told by a doctor or other health worker that they have raised blood pressure [hypertension] or raised blood glucose [diabetes]* | | | |
| **Treatment: lifestyle counseling** | | | |
| Counseled to exercise | X | X | X |
| Counseled to maintain a healthy body weight or lose weight | X | X | X |
| Counseled to reduce salt intake | X | | X |
| **Treatment: pharmacological therapy use[†]** | | | |
| Antihypertensive | X | | X |
| Glucose-lowering | | X | X |
| *Oral glucose-lowering medication or insulin* | | | |
| Antihypertensive & Glucose lowering | | | X |
| Cholesterol-lowering (statin) | | X | X |
| **Management targets** | | | |
| Blood pressure (BP) | X | | X |
| *Individuals with systolic BP <140 mmHg & diastolic BP <90 mmHg* | | | |
| Blood glucose (BG) | | X | X |
| *Individuals with HbA1c <7.0% or FPG <7.0 mmol/L (<126 mg/dl) if HbA1c not available* | | | |
| BP & BG | | | X |
| BG & statin use | | X | X |
| BP, BG, & statin use | | | X |

*Abbreviations*: FPG, fasting plasma glucose; HbA1c, hemoglobin A1c; mg/dl, milligram/deciliter; mmHg, millimeter of mercury; mmol/L, millimole/liter.

[a]All indicators were assessed in the 55 countries included in this analysis, except for the lifestyle counseling indicators (33–34 countries) and indicators that include status use (34 countries).

[b]Individuals with *hypertension only* were those with blood pressure ≥140/90 mmHg or self-reported antihypertensive medication use or self-reported diagnosis by a clinician, without diabetes. Individuals with *diabetes only* were those with a FPG of 7.0 mmol/L (126 mg/dL) or above, random plasma glucose 11.1 mmol/L (200mg/dL) or above, an HbA1c measurement of 6.5% or above or self-reporting using glucose-lowering medications or self-reported diagnosis by a clinician, without hypertension. Individuals with *hypertension and diabetes (hypertension-diabetes)* were those with concurrent hypertension and diabetes, and they were our primary study population.

## Statistical analysis

For each group of interest, we estimated the percentage of individuals who received healthcare services and achieved condition management overall and stratified by survey implementation-year groups (2009–2014 and 2015–2019) and World Bank income group. We combined low-income (*n* = 11) and lower-middle-income (*n* = 26) countries because of data sparsity and reported stratified results as low/lower-income countries (L-MICs) and upper-middle-income countries (UMICs).

In all analyses, we accounted for the complex survey design by adjusting for stratification and clustering at the primary sampling unit (PSU) using the 'srvyr' R package [34]. Additionally, we used sampling weights adjusting for selection probability, nonresponse, and differences between the sample and target population. The main interest of our analysis is at the health system level; therefore, we rescaled survey weights to ensure equal contribution of each survey. If survey weights were missing but biomarker information was available, the country-average weight was assigned. For all other data, we did not replace or impute missing values.

All models also include countries as indicator variables. We included only individuals with all relevant covariate and indicator data for each outcome analysis. Analyses were done in R version 4.1.2 [35].

### Exploratory analysis

In addition to the risk factor approach (hypertension and diabetes), we also considered the total risk approach, the preferred strategy [36] to identify those at high risk of debilitating (e.g., heart attacks, strokes, etc.,) and fatal CVD outcomes. The total risk approach considers several risk factors including age, biological sex, body mass index (BMI), tobacco use, diabetes diagnosis, BP, and blood cholesterol to calculate a 10-year CVD risk score [37]. We estimated the percentage of individuals who received healthcare services and achieved appreciate management goals in the hypertension-diabetes group among those with 10-year CVD risk score <10% and ≥10%. We used the 2019 WHO office-based risk equations to estimate 10-year CVD risk [37].

### Ethics

The institutional review board at the University of North Carolina at Chapel Hill approved this study as exempt because of the use of de-identified data (Exemption # 21–2860).

### Results

Of the 55 included countries, 37 were L-MICs and 19 UMICs (**Table 2**) and the average response rate was 86.7%. The pooled sample included 280,783 non-pregnant individuals. Of those, 90,086 (38.8% of the population-weighted sample) lived with hypertension only, 11,975 (4.9%) lived with diabetes only, and 16,228 (9.3%) lived with hypertension-diabetes (**Table 3**). The hypertension-diabetes group had the highest weighted percentage of individuals who were aged 60–69 years (30.1%), women (57.3%), and had a BMI ≥ 30 kg/m$^2$ (47.3%); however, they also had the lowest weighted percentage of individuals who were current tobacco smokers (15.3%). The share of missing values for each characteristic and outcome are provided by country in **S6** and **S7 Tables**, respectively.

In **Fig 1** we show the percentage of individuals who received healthcare services and achieved appropriate management across the three study groups in the pooled country sample. Among the hypertension-diabetes group, 64.1% (95% CI: 61.8–66.4) were diagnosed with both conditions, which was more than sixteen percentage points higher than those diagnosed in the hypertension only (47.4% [95% CI: 45.3–49.6]) and diabetes only (46.7% [95% CI: 44.1–49.2]) groups (**Fig 1A**). Across the three lifestyle counseling indicators, the hypertension-diabetes group reported a higher uptake of lifestyle counseling than the hypertension only and diabetes only groups. For example, more than half of those in the hypertension-diabetes group (55.8% [95% CI: 52.1–59.5]) reported receiving exercise counseling compared to 39.7% (95% CI: 38.3–41.0]) and 42.3% (95% CI: 38.9–45.7) of those in the hypertension only and diabetes only groups, respectively (**Fig 1B**).

Medication use was low in all three groups with 20–30% taking treatment, and this was no different across groups (**Fig 1C**). When disaggregating the combined pharmacological therapy indicator in the hypertension-diabetes group, no individual condition was more likely to be treated (38.3% [95% CI: 34.8–41.8] using antihypertensive and 42.3% [95% CI: 39.4–45.2] using glucose-lowering medications); however, the treatment usage was higher for each of the individual conditions than both conditions (24.6% [95% CI: 22.1–27.2]). Statin use was 9.5% (95% CI: 8.3–10.7) in the hypertension-diabetes group, which was more than double that of the statin use in the diabetes only group (4.6% [95% CI: 3.5–5.6]).

**Table 2. Survey characteristics.**

| Geographic region and country | Income group | Year | Response rate, (%) | N, after exclusion[a] | Female, (%) | Median age, years (IQR) |
|---|---|---|---|---|---|---|
| *East, South, and Southeast Asia* | | | | | | |
| Bangladesh | L-MIC | 2018 | 83.3 | 3,235 | 48.3 | 49 (44–56) |
| Bhutan | L-MIC | 2014 | 96.9 | 1,307 | 57.6 | 50 (45–57) |
| Cambodia | L-MIC | 2010 | 96.3 | 3,077 | 65.4 | 50 (45–56) |
| India | L-MIC | 2015–16 | 97.6 | 175,222 | 82.6 | 45 (42–47) |
| Indonesia | L-MIC | 2014 | 90.5 | 2,719 | 57.8 | 56 (47–61) |
| Laos | L-MIC | 2013 | 99.2 | 1,162 | 58.0 | 49 (44–55) |
| Myanmar | L-MIC | 2014 | 94.0 | 5,219 | 65.5 | 51 (45–57) |
| Nepal | L-MIC | 2019 | 86.4 | 2,468 | 58.8 | 51 (45–60) |
| Sri Lanka | L-MIC | 2014 | 72.0 | 2,586 | 59.8 | 53 (46–60) |
| Timor-Leste | L-MIC | 2014 | 96.3 | 1,211 | 52.8 | 51 (44–61) |
| Vietnam | L-MIC | 2015 | 97.4 | 1,842 | 56.9 | 52 (45–59) |
| *Europe and Central Asia* | | | | | | |
| Azerbaijan | UMIC | 2017 | 97.3 | 1,694 | 60.0 | 55 (48–60) |
| Belarus | UMIC | 2016 | 87.1 | 3,288 | 59.7 | 54 (47–61) |
| Georgia | L-MIC | 2016 | 75.7 | 2,297 | 73.0 | 56 (49–63) |
| Kyrgyzstan | L-MIC | 2013 | 100.0 | 1,556 | 63.2 | 51 (46–57) |
| Moldova | L-MIC | 2013 | 83.5 | 2,404 | 63.9 | 55 (49–62) |
| Mongolia | L-MIC | 2013 | 97.4 | 971 | 57.4 | 49 (44–54) |
| Romania | UMIC | 2015–16 | 69.1 | 1,017 | 53.5 | 54 (46–62) |
| Tajikistan | L-MIC | 2016 | 94.0 | 1,258 | 56.8 | 50 (45–57) |
| *Latin America and the Caribbean* | | | | | | |
| Chile | UMIC | 2009–10 | 85.0 | 2,345 | 60.2 | 53 (46–60) |
| Costa Rica | UMIC | 2010 | 87.8 | 1,452 | 75.1 | 52 (46–60) |
| Ecuador | UMIC | 2018 | 69.4 | 2,040 | 56.9 | 52 (46–60) |
| Guyana | UMIC | 2016 | 77.0 | 442 | 61.1 | 52 (46–59) |
| Mexico | UMIC | 2009–12 | 90.0 | 4,868 | 54.9 | 54 (48–60) |
| St. Vincent & the Grenadines | UMIC | 2013 | 67.8 | 594 | 57.4 | 52 (46–59) |
| *Middle East and North Africa* | | | | | | |
| Algeria | UMIC | 2016 | 93.8 | 3,131 | 54.2 | 50 (44–58) |
| Iran | UMIC | 2016 | 98.4 | 10,309 | 54.1 | 52 (45–59) |
| Iraq | UMIC | 2015 | 98.8 | 1,730 | 58.7 | 50 (44–59) |
| Lebanon | UMIC | 2017 | 65.9 | 790 | 62.7 | 52 (47–58) |
| Morocco | L-MIC | 2017 | 89.0 | 2,484 | 62.9 | 52 (46–60) |
| *Oceania* | | | | | | |
| Kiribati | L-MIC | 2015 | 55.0 | 528 | 54.5 | 50 (45–57) |
| Marshall Islands | UMIC | 2017 | 92.3 | 1,146 | 50.0 | 50 (44–57) |
| Samoa | L-MIC | 2013 | 64.0 | 702 | 62.0 | 50 (45–56) |
| Solomon Islands | L-MIC | 2015 | 58.4 | 824 | 50.2 | 50 (45–58) |
| Tuvalu | UMIC | 2015 | 76.0 | 545 | 55.6 | 54 (47–59) |
| Vanuatu | L-MIC | 2011 | 94.0 | 2,262 | 47.1 | 50 (44–56) |
| *Sub-Saharan Africa* | | | | | | |
| Benin | L-MIC | 2015 | 98.6 | 1,978 | 48.5 | 50 (44–56) |
| Botswana | UMIC | 2014 | 63.0 | 1,280 | 70.7 | 51 (45–58) |
| Burkina Faso | L-MIC | 2013 | 99.1 | 1,697 | 48.0 | 49 (44–55) |
| Comoros | L-MIC | 2011 | 96.5 | 1,233 | 72.2 | 50 (44–57) |
| Eritrea | L-MIC | 2010 | 97.0 | 2,933 | 65.5 | 51 (45–60) |

*(Continued)*

**Table 2.** (Continued)

| Geographic region and country | Income group | Year | Response rate, (%) | N, after exclusion[a] | Female, (%) | Median age, years (IQR) |
|---|---|---|---|---|---|---|
| Eswatini | L-MIC | 2014 | 76.0 | 1,025 | 68.3 | 52 (45–60) |
| Kenya | L-MIC | 2015 | 93.0 | 1,601 | 59.3 | 51 (44–59) |
| Lesotho | L-MIC | 2012 | 80.0 | 1,116 | 67.7 | 52 (46–59) |
| Liberia | L-MIC | 2011 | 87.1 | 534 | 50.2 | 48 (43–55) |
| Namibia | UMIC | 2013 | 95.8 | 2,464 | 58.4 | 49 (44–55) |
| Rwanda | L-MIC | 2012 | 99.8 | 2,270 | 64.4 | 49 (44–56) |
| São Tomé and Principe | L-MIC | 2009 | 95.0 | 897 | 61.9 | 49 (44–55) |
| Seychelles | UMIC | 2013 | 73.0 | 851 | 56.2 | 52 (46–57) |
| Sudan | L-MIC | 2016 | 95.0 | 2,909 | 57.9 | 50 (45–58) |
| Tanzania | L-MIC | 2012 | 94.7 | 2,499 | 50.5 | 49 (44–56) |
| Togo | L-MIC | 2010 | 91.0 | 1,170 | 48.2 | 48 (44–55) |
| Uganda | L-MIC | 2014 | 92.2 | 1,185 | 60.3 | 50 (44–57) |
| Zambia | L-MIC | 2017 | 74.0 | 1,280 | 63.5 | 50 (44–59) |
| Zanzibar | L-MIC | 2011 | 91.0 | 1,136 | 58.5 | 49 (45–55) |
| *Global* | | 2009–19 | 86.7 | 280,783 | 73.6 | 45 (42–49) |

Abbreviation: L-MIC: Low/lower-income countries; UMIC: Upper-middle-income countries.

[a]Individuals aged 25–69 years old with non-missing diabetes biomarkers and blood pressure measurements who reported not being pregnant were included.

The percentage of individuals who achieved appropriate management targets was lower among the hypertension-diabetes group (6.6% [95% CI: 5.4–7.8]) than the hypertension only (17.6% [95% CI: 16.4–18.8]) and diabetes only (13.3% [95% CI: 10.7–15.8]) groups (**Fig 1D**).

**Table 3. Individual characteristics of overall sample[a].**

| | Total Pooled Sample | | Hypertension Only | | Diabetes Only | | Hypertension-Diabetes | |
|---|---|---|---|---|---|---|---|---|
| Characteristic | Unweighted, N | Weighted, % | Unweighted, N | Weighted, % | Unweighted, N | Weighted, % | Unweighted, N | Weighted, % |
| **Overall**[b] | 280,783 | | 89,906 | | 11,958 | | 16,210 | |
| *Female* | 205,591 | 52.8 | 64,655 | 54.1 | 7,979 | 53.2 | 10,908 | 57.5 |
| *Age (years)* | | | | | | | | |
| 40–49 | 211,925 | 48.0 | 60,933 | 41.0 | 8,171 | 45.3 | 7,979 | 28.6 |
| 50–59 | 45,559 | 33.9 | 17,867 | 36.8 | 2,557 | 36.6 | 4,579 | 41.4 |
| 60–69 | 23,300 | 18.1 | 11,106 | 22.2 | 1,230 | 18.1 | 3,652 | 30.1 |
| *Education* | | | | | | | | |
| No schooling | 107,461 | 22.2 | 30,657 | 21.4 | 3,348 | 15.3 | 3,905 | 16.0 |
| Primary | 66,361 | 36.9 | 22,563 | 36.7 | 3,189 | 37.1 | 4,606 | 35.9 |
| Secondary or higher | 105,962 | 40.9 | 36,338 | 41.9 | 5,337 | 47.6 | 7,552 | 48.1 |
| *BMI (kg/m²)* | | | | | | | | |
| <18.5 | 31,744 | 6.6 | 6,426 | 4.5 | 811 | 3.6 | 355 | 1.3 |
| 18.5–<25.0 | 140,762 | 40.3 | 39,084 | 34.7 | 4,798 | 31.6 | 4,226 | 19.8 |
| 25.0–<30.0 | 69,850 | 28.8 | 27,025 | 30.9 | 3,864 | 33.5 | 5,742 | 31.3 |
| ≥30.0 | 36,962 | 24.3 | 16,875 | 29.9 | 2,353 | 31.3 | 5,684 | 47.6 |
| *Current tobacco smoker* | 62,926 | 20.3 | 18,517 | 18.0 | 2,509 | 21.1 | 2,555 | 15.2 |

[a] Individuals aged -69 years old with non-missing diabetes biomarkers and blood pressure measurements who reported not being pregnant were included. Proportions are calculated using weights provided by the individual surveys, readjusted such that each country is weighted equally.

[b]Among the total pooled sample, 89,906 (38.9% of the population-weighted sample) lived with hypertension only, 11,958 (4.9%) lived with diabetes only, and 16,210 (9.4%) lived with both hypertension and diabetes (hypertension-diabetes).

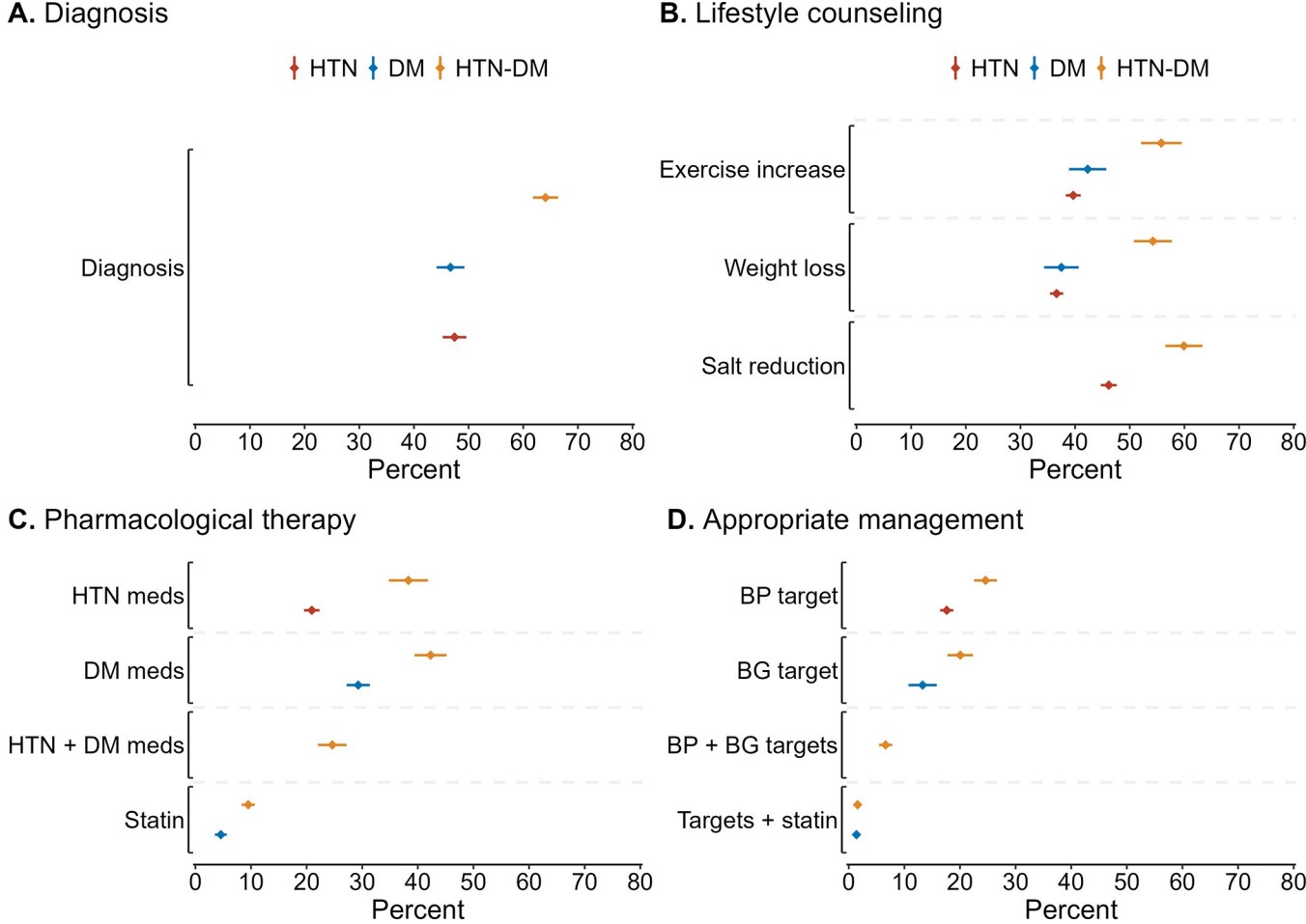

**Fig 1.** **(A)** The percentage of individuals who were aware of their condition (diagnosed), **(B)** received lifestyle counseling, **(C)** used pharmacological therapy and **(D)** achieved treatment control targets for cardiovascular disease risk reduction among those with hypertension only, diabetes only, and hypertension-diabetes in 55 low- and middle-income countries. *Abbreviations*: BG, **blood** glucose; BP, blood pressure; CVD, cardiovascular disease; DM, diabetes mellitus; HTN, hypertension; meds, medications.

When accounting for statin use as part of appropriate management in the 34 countries where these data were available, a substantially smaller share of the hypertension-diabetes (1.6% [95% CI: 1.1–2.0]) and diabetes only (1.4% ([95% CI: 0.8–1.9]) groups achieved targets (**Fig 1D**).

When stratified by World Bank income group, the percentage of individuals who received healthcare services and achieved appropriate management targets was higher in upper-middle-income than low- or lower-middle-income countries for all three groups (**Fig 2**). We did not observe statistically significant differences in outcomes (diagnosis, treatment, and appropriate management) between surveys conducted during 2009–2014 and 2015–2019 (**S9 Table**).

In our exploratory analysis conducted among the hypertension-diabetes group, most (71.5% [95% CI: 69.5, 73.4]) had an estimated predicted CVD risk score ≥10% (**S8 Table**). There were no differences in diagnosis awareness and receipt of lifestyle counseling outcomes between those with a CVD risk score <10% versus ≥10% (**Fig 3**). Those with CVD risk score ≥10% (24.4% [95% CI: 21.7–27.1]) were slightly more likely to report using antihypertensive and glucose-lowering medications than those with CVD risk score <10% (18.4% [95% CI: 15.1–21.6]). However, those with CVD risk score ≥10% (4.7% [95% CI: 3.4–5.9]) were less

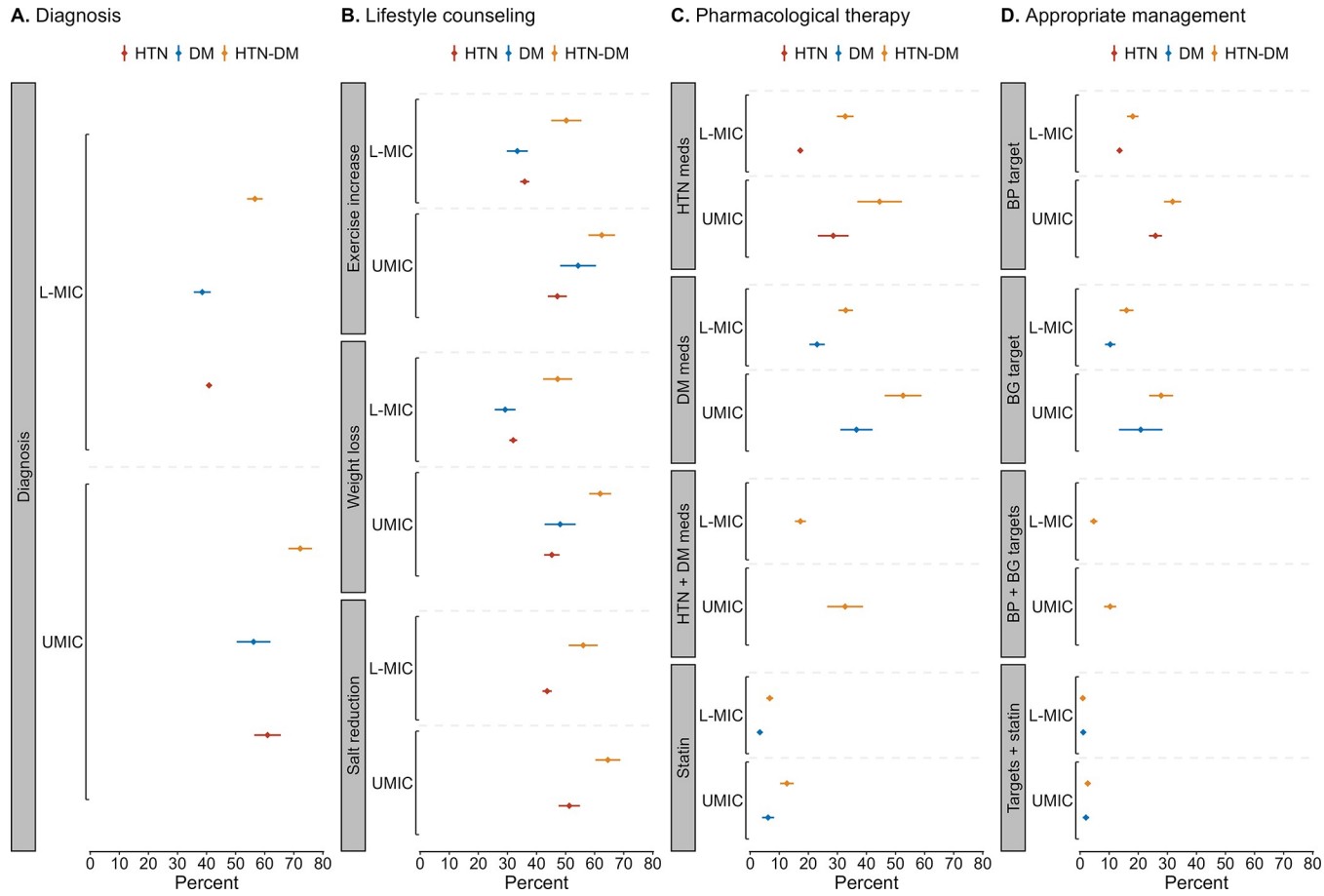

**Fig 2.** **(A)** The percentage of individuals who were aware of their condition (diagnosed), **(B)** received lifestyle counseling, **(C)** used pharmacological therapy, and **(D)** achieved treatment control targets for cardiovascular disease risk reduction among those with hypertension only, diabetes only, and hypertension-diabetes in 55 low- and middle-income countries, by World Bank income group. *Abbreviations*: BG, blood glucose; BP, blood pressure; CVD, cardiovascular disease; DM, diabetes mellitus; HTN, hypertension; meds, medications; L-MIC, low/lower middle-income countries; UMIC, upper-middle-income countries.

likely to achieve the appropriate hypertension and diabetes management targets than those with CVD risk score <10% (13.8% [95% CI: 9.1–18.5]).

## Discussion

Our study of nationally representative, individual-level data from 55 LMICs shows that health systems in LMICs are reaching a larger share of people living with both hypertension and diabetes—those with highest risk of poor CVD-related health outcomes—compared to those with either hypertension or diabetes alone. However, we also found substantial unmet need for all forms of clinical care in people with both risk factors. Only one in four people with both conditions used medications to address these risk factors. Additionally, less than 10% achieved both BP and BG management targets, and this figure dropped to 1.6% after accounting for statin use, which was low overall (5% in the diabetes-only and 10% in the hypertension-diabetes group). Finally, fewer than one in five people in each of the three study groups achieved appropriate risk factor management targets. Findings suggest an urgent need to implement cost-effective population-level interventions that shift the clinical care paradigm from disease-specific care to comprehensive CVD care for all three groups, especially for those with multiple CVD risk factors [38].

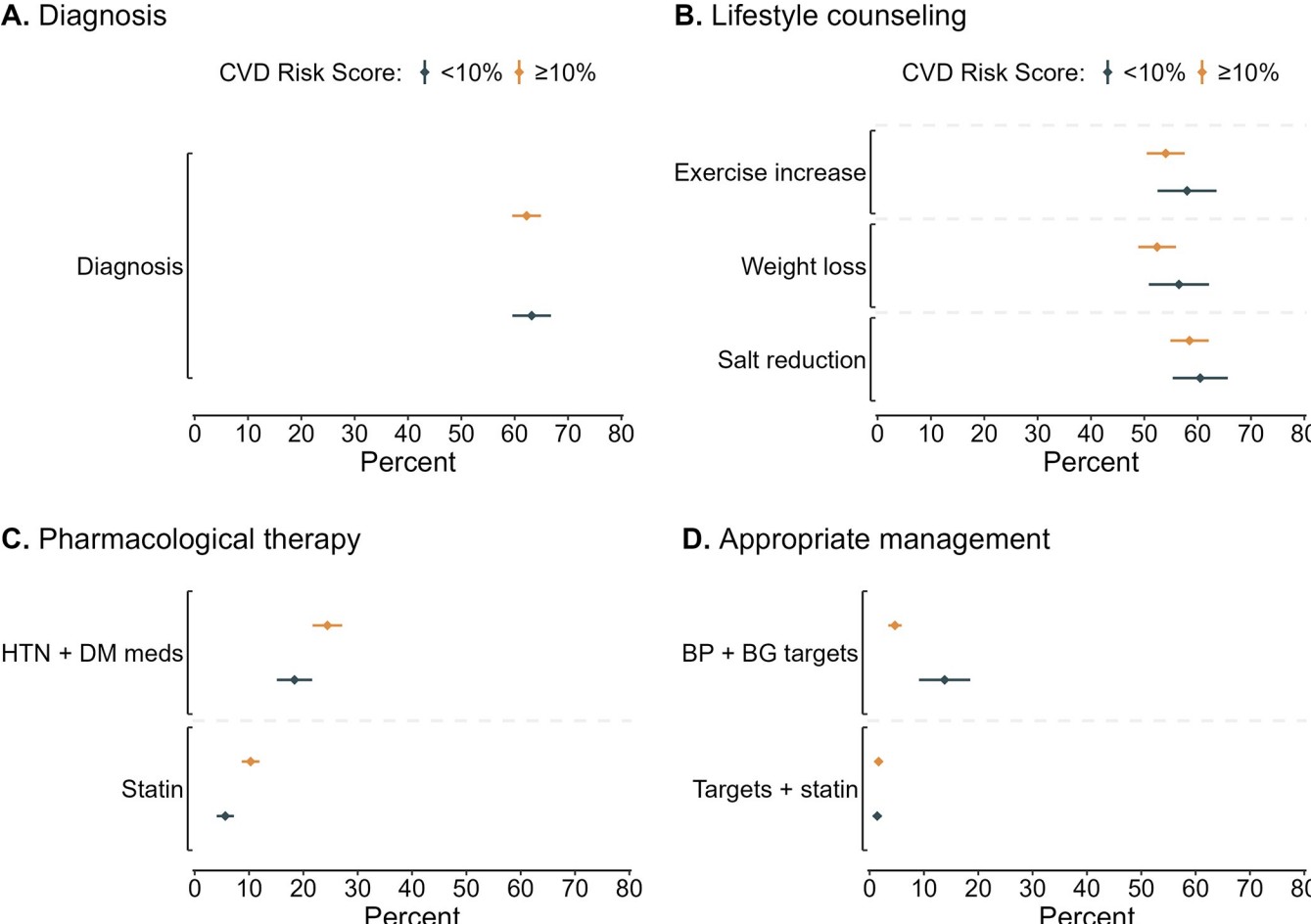

**Fig 3. (A)** The percentage of individuals who were aware of their condition (diagnosed), **(B)** received lifestyle counseling, **(C)** used pharmacological therapy, and **(D)** achieved treatment control targets for cardiovascular disease risk reduction among those with hypertension-diabetes 10-year predicted CVD risk score <10% versus ≥10% in 55 low- and middle-income countries[a]. *Abbreviations*: BG, blood glucose; BP, blood pressure; CVD, cardiovascular disease; DM, diabetes mellitus; HTN, hypertension; meds, medications. [a]We used 2019 WHO office-based risk equations to estimate 10-year CVD risk [37].

Evidence from LMICs suggests that individuals with multiple conditions, including hypertension and diabetes, have more interactions with the health system [13, 39]. The coexistence of these conditions exacerbates poorer health status, requiring individuals to seek both primary and secondary preventative care, while simultaneously providing the health system with more opportunities to diagnose and deliver care [6]. Previous studies in LMICs have quantified the number of interactions and cost of healthcare utilization among those with multiple CVD risk factors [13, 38, 40]. Our study expands upon this literature by showing that the share of people achieving appropriate BP and BG management targets is low in individuals with hypertension only, diabetes only, and hypertension-diabetes, despite greater receipt of healthcare services among those with both conditions.

Among the hypertension-diabetes group, a greater proportion reported receiving lifestyle counseling than pharmacological therapies (e.g., increase exercise: 56% versus concurrent anti-hypertensive and glucose-lowering medication use: 25%). This may be because of reduced access to pharmacological therapies or low health literacy. Additionally, those reporting receipt of lifestyle counseling may be a younger, leaner population in whom HbA1c (or equivalent FPG) is <8% or BP <150/95 mmHg and their clinicians are willing to counsel on lifestyle

change as recommended by country-specific and leading international guidelines [28, 33] for risk factor management [41]. Translating this evidence into health policies and clinical practice to achieve health gains is particularly challenging in LMICs because of resource constraints, prioritization, and complex societal factors such as the increasing availability and consumption of inexpensive processed foods [42–44].

Reduced availability and affordability of pharmacological therapies for CVD risk management and care in LMICs, especially when multiple treatments are required, are key contributors to the low proportion of medication use among those with multiple CVD risk factors [45, 46]. Given the effectiveness of pharmacological therapies including antihypertensive, glucose-lowering, and statin medications on CVD-related outcomes, this lack of access is consequential and associated with a higher risk of adverse health outcomes, including stroke and mortality in LMICs. Among individuals with hypertension-diabetes, medication use for one condition was higher than for both conditions. This could be driven by differential availability and affordability of antihypertensive and glucose-lowering medications [45, 46] or clinical prioritization due to symptomatic severity of one condition over the other [47] when resources are limited. Here, we found greater use of pharmacological therapy for one condition in individuals with hypertension-diabetes compared to those who had either condition alone, which suggests that people with both conditions are interacting with the health system and provided more opportunities to receive needed care. Approaches to improve access to and use of pharmacological therapies among those with multiple CVD risk factors in LMICs, including the procurement of quality-assured medications that are affordable to the greater public, should be considered and implemented [45].

Our finding that less than 10% of individuals with hypertension-diabetes achieved BP and BG management targets reflects the reality that the coexistence of these CVD risk factors dramatically increases the complexity of managing individuals' symptoms to prevent further deterioration of health and quality of life [5, 6]. However, we found that more individuals in the hypertension-diabetes group achieved control of at least one target than those in either the hypertension only or diabetes only groups. When comparing the achievement of the two targets, we noted that achieving the BG target (diabetes only: 13% and hypertension-diabetes: 20%) was less common than achieving the BP target (hypertension only: 18% and hypertension-diabetes: 25%). A recent study conducted in the UK concurs with our finding that the coexistence of multiple CVD risk factors was associated with a higher probability of achieving appropriate management [48] while others have found that individuals with hypertension-diabetes were less likely to achieve BP control [49, 50]. Nevertheless, the levels of risk factor management in all three groups in our study were lackluster, suggesting an urgent need for improvements in CVD care in LMICs.

Our study has limitations. First, our definitions of hypertension and diabetes were limited to a single time point or measurement which might have resulted in the misclassification of our three study groups. In clinical practice, hypertension management is based on BP measured during multiple consecutive healthcare visits. However, we used three BP measurements from a single occasion to diagnose hypertension [51–53], and higher thresholds for hypertension definition compared to current guidelines that use a BP >130/80 mmHg. For diabetes, most surveys collected a single capillary glucose measurement, which other studies have found to under- or over-diagnose diabetes [54]. However, biological measurements captured on a single time point are commonly used in high-quality population-based surveys to estimate disease prevalence and evaluate health system performance [55]. Additionally, we included reported diagnosis awareness and medication use in our definition to identify individuals with BP and diabetes biomarker measurements below the diagnostic thresholds who may have benefited from lifestyle counseling and pharmacological therapies and, therefore, would not be

detected based on biological measurements alone. Second, we used self-reported information to define our diagnosis awareness, lifestyle counseling, and pharmacological therapy outcomes which might lead to differential misclassification of our outcomes due to recall bias. It should be noted that prior studies have found high accuracy of self-reported CVD histories and medications [56, 57]. Third, we used data collected from surveys conducted in different countries over ten years. Although most surveys (84%) used the WHO STEPS questionnaire and similar approaches to measure BP and diabetes biomarkers, differences in the translation and phrasing of questionnaires and implementation of biological measurement procedures might have contributed to variations in our estimates. Finally, the 55 LMICs included in this study may not represent all LMICs globally and encompass a heterogeneous group of countries with different health systems. However, including them together in analyses allows us to better understand the state of CVD care for individuals at the highest risk of developing CVD-related disability and mortality in low-resource settings.

In conclusion, we found that individuals with concurrent hypertension and diabetes were more likely to have been diagnosed and treated for CVD risk factors than individuals with only one of these conditions. However, using the recently adopted global coverage targets for diabetes by 2030 as a benchmark [58], there remains a substantial gap in appropriate CVD risk factor management as less than one in ten people with concurrent risk factors achieved both BP and BG targets, and even fewer achieved these metrics along with statin use. To achieve these targets, policy efforts and investments should increase health systems' capacity to deliver cost-effective population-level interventions and shift the clinical care paradigm from disease-specific care to comprehensive care.

## Supporting information

**S1 Table. Detailed definitions the blood pressure and diabetes biomarker measurements.**
(DOCX)

**S2 Table. Blood pressure measurement details.**
(DOCX)

**S3 Table. Diabetes biomarker measurement details.**
(DOCX)

**S4 Table. Lipid biomarker measurement details.**
(DOCX)

**S5 Table. Select questions from generic STEPS surveys.**
(DOCX)

**S6 Table. Number and percent of participants with missing predictor variables by country.**
(DOCX)

**S7 Table. Number and percent of participants with missing outcome indicator variables by country.**
(DOCX)

**S8 Table. Distribution of 10-year predicted cardiovascular disease (CVD) risk score among individuals with hypertension only, diabetes only and hypertension and diabetes (hypertension-diabetes).**
(DOCX)

**S9 Table. The proportion of individuals who were aware of their condition (diagnosed), had received lifestyle counseling, used pharmacological therapy, and achieved appropriate**

**management for cardiovascular disease risk reduction among those with hypertension (HTN) only, diabetes (DM) only, and hypertension and diabetes (HTN-DM) in 55 low- and middle-income countries overall and by world bank income group and survey year groups (2009–2014 and 2015–2019).**
(DOCX)

## Acknowledgments

We would like to thank the survey teams and survey participants for taking part in this research. Funding to support this analysis was provided by the Harvard T.H. Chan School of Public Health McLennan Fund: Dean's Challenge Grant Program.

## Author Contributions

**Conceptualization:** Alpha Oumar Diallo, David Flood, Emily W. Gower, Jennifer Manne-Goehler.

**Data curation:** Alpha Oumar Diallo, Maja E. Marcus, David Flood, Michaela Theilmann, Nicholas E. Rahim, Alan Kinlaw, Nora Franceschini, Til Stürmer, Mohsen Abbasi-Kangevari, Kokou Agoudavi, Glennis Andall-Brereton, Krishna Aryal, Silver Bahendeka, Brice Bicaba, Pascal Bovet, Maria Dorobantu, Farshad Farzadfar, Seyyed-Hadi Ghamari, Gladwell Gathecha, David Guwatudde, Mongal Gurung, Corine Houehanou, Dismand Houinato, Nahla Hwalla, Jutta Jorgensen, Gibson Kagaruki, Khem Karki, Joao Martins, Mary Mayige, Roy Wong McClure, Sahar Saeedi Moghaddam, Omar Mwalim, Kibachio Joseph Mwangi, Bolormaa Norov, Sarah Quesnel-Crooks, Abla Sibai, Lela Sturua, Lindiwe Tsabedze, Chea Wesseh, Pascal Geldsetzer, Rifat Atun, Sebastian Vollmer, Till Bärnighausen, Justine Davies, Mohammed K. Ali, Jacqueline A. Seiglie, Emily W. Gower, Jennifer Manne-Goehler.

**Formal analysis:** Alpha Oumar Diallo, Maja E. Marcus, Emily W. Gower, Jennifer Manne-Goehler.

**Writing – original draft:** Alpha Oumar Diallo, Maja E. Marcus, David Flood, Dessie V. Tien, Emily W. Gower, Jennifer Manne-Goehler.

**Writing – review & editing:** Alpha Oumar Diallo, Maja E. Marcus, David Flood, Michaela Theilmann, Nicholas E. Rahim, Alan Kinlaw, Nora Franceschini, Til Stürmer, Dessie V. Tien, Mohsen Abbasi-Kangevari, Kokou Agoudavi, Glennis Andall-Brereton, Krishna Aryal, Silver Bahendeka, Brice Bicaba, Pascal Bovet, Maria Dorobantu, Farshad Farzadfar, Seyyed-Hadi Ghamari, Gladwell Gathecha, David Guwatudde, Mongal Gurung, Corine Houehanou, Dismand Houinato, Nahla Hwalla, Jutta Jorgensen, Gibson Kagaruki, Khem Karki, Joao Martins, Mary Mayige, Roy Wong McClure, Sahar Saeedi Moghaddam, Omar Mwalim, Kibachio Joseph Mwangi, Bolormaa Norov, Sarah Quesnel-Crooks, Abla Sibai, Lela Sturua, Lindiwe Tsabedze, Chea Wesseh, Pascal Geldsetzer, Rifat Atun, Sebastian Vollmer, Till Bärnighausen, Justine Davies, Mohammed K. Ali, Emily W. Gower, Jennifer Manne-Goehler.

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
