## [Decision Letter · Decision Letter 0]

21 Aug 2023

PGPH-D-23-00340

Multiple cardiovascular risk factor care in 56 low- and middle-income countries: a cross-sectional analysis of nationally representative, individual-level data from 281,322 adults

Dear Dr. Gower,

Thank you for submitting your manuscript to PLOS Global Public Health. After careful consideration, we feel that it has merit but does not fully meet PLOS Global Public Health’s publication criteria as it currently stands. Therefore, we invite you to submit a revised version of the manuscript that addresses the points raised during the review process.

We look forward to receiving your revised manuscript.

Kind regards,

Ziyue Wang, M.D.

Guest Editor

Reviewers' comments:

Reviewer's Responses to Questions

**Comments to the Author**

1. Does this manuscript meet PLOS Global Public Health’s publication criteria? Is the manuscript technically sound, and do the data support the conclusions? The manuscript must describe methodologically and ethically rigorous research with conclusions that are appropriately drawn based on the data presented.

Reviewer #1: Yes

Reviewer #2: Yes

2. Has the statistical analysis been performed appropriately and rigorously?

Reviewer #1: Yes

Reviewer #2: Yes

3. Have the authors made all data underlying the findings in their manuscript fully available (please refer to the Data Availability Statement at the start of the manuscript PDF file)?

Reviewer #1: Yes

Reviewer #2: Yes

4. Is the manuscript presented in an intelligible fashion and written in standard English?

Reviewer #1: Yes

Reviewer #2: Yes

5. Review Comments to the Author

Reviewer #1: This paper uses nationally representative data from 56 LMICs to measure care cascade steps among individuals with hypertension only, diabetes only, and hypertension and diabetes both. The paper is well-written and easy to follow and I do not have any concerns about the validity of the statistical analyses.

My primary concern is around the framing and scientific and policy relevance of the paper. Papers documenting the hypertension cascade (regardless of diabetes) and the diabetes cascade (regardless of hypertension) are valuable for revealing gaps and focusing policy attention. However, the main contribution of this paper is the focus on those that are both hypertensive and diabetic. It isn't clear to me what additional insights we gain from the authors' analysis of this population that we wouldn't have gotten from focusing on just those with hypertension or just those with diabetes. Put differently, if there are existing policies focused on hypertension improvement and diabetes improvement, wouldn't these also catch those with both conditions? I struggle to think of how we might especially miss those with both conditions.

One suggestion to the authors for increasing the relevance of the paper might be to focus on "missed opportunities" for diagnosis or treatment. For example, what share of those with both hypertension and diabetes are diagnosed or taking treatment for just one of the two conditions (e.g. what proportion of those with both are diagnosed for hypertension but not diabetes or taking treatment for diabetes but not hypertension)? Such discrepancies - if they exist in the data - have more direct policy relevance as they suggest that policies and care focused purely on hypertension are missing out on the opportunity to diagnose and treat a significant share of people with diabetes (and potentially vice versa). To play devil's advocate, if most people with hypertension don't have diabetes, then maybe it is actually more cost-effective to just focus on hypertension (hopefully your data shows that this isn't the case).

My second concern with the paper is around the multivariable prediction analyses. As it is written, this set of analyses isn't motivated and seems almost mechanical, as if it is something everyone has to do in papers (I am personally guilty of this as well). I would encourage the authors to motivate these analyses better and tailor the models to the motivation. If the goal is to identify disparities, then simple stratified estimates/bivariate regressions would be the appropriate way (since the real world is not adjusted). If the goal is to develop a prediction model that could be used by practitioners, then the authors might consider predictive accuracy as the goal and compare different model approaches to predicting. As it stands, a multivariable regression of different social and demographic variables does satisfyingly answer either motivation.

Reviewer #2: 1. How to ensure national representativeness when different countries have different population sizes and different sample sizes included in the analysis, for example, China, where just over 500 participants were included in the analysis.

2. The surveys included in the study span a relatively long period of time, how to eliminate the time factor from interfering with the results?

6. PLOS authors have the option to publish the peer review history of their article (what does this mean?). If published, this will include your full peer review and any attached files.

**Do you want your identity to be public for this peer review?** For information about this choice, including consent withdrawal, please see our Privacy Policy.

Reviewer #1: No

Reviewer #2: No

---

## [Decision Letter · Decision Letter 1]

23 Feb 2024

Multiple Cardiovascular Risk Factor Care in 55 Low- and Middle-Income Countries: A Cross-Sectional Analysis of Nationally Representative, Individual-Level Data from 280,783 adults

PGPH-D-23-00340R1

Dear Dr Gower,

We are pleased to inform you that your manuscript 'Multiple Cardiovascular Risk Factor Care in 55 Low- and Middle-Income Countries: A Cross-Sectional Analysis of Nationally Representative, Individual-Level Data from 280,783 adults' has been provisionally accepted for publication in PLOS Global Public Health.

**Please note that your manuscript will not be scheduled for publication until you have made the required changes to address the final issues raised by the reviewers**, so a swift response is appreciated.

Best regards,

Ziyue Wang, PhD

Guest Editor

Reviewer Comments (if any, and for reference):

Reviewer's Responses to Questions

**Comments to the Author**

1. If the authors have adequately addressed your comments raised in a previous round of review and you feel that this manuscript is now acceptable for publication, you may indicate that here to bypass the “Comments to the Author” section, enter your conflict of interest statement in the “Confidential to Editor” section, and submit your "Accept" recommendation.

Reviewer #1: (No Response)

Reviewer #2: All comments have been addressed

2. Does this manuscript meet PLOS Global Public Health’s publication criteria? Is the manuscript technically sound, and do the data support the conclusions? The manuscript must describe methodologically and ethically rigorous research with conclusions that are appropriately drawn based on the data presented.

Reviewer #1: Yes

Reviewer #2: Yes

3. Has the statistical analysis been performed appropriately and rigorously?

Reviewer #1: Yes

Reviewer #2: Yes

4. Have the authors made all data underlying the findings in their manuscript fully available (please refer to the Data Availability Statement at the start of the manuscript PDF file)?

Reviewer #1: Yes

Reviewer #2: Yes

5. Is the manuscript presented in an intelligible fashion and written in standard English?

Reviewer #1: Yes

Reviewer #2: Yes

6. Review Comments to the Author

Reviewer #1: I thank the reviewers for taking my comments seriously and updating the manuscript. I have no further major comments but a few minor comments that may help to better motivate the urgency of the paper:

1. The analyses by total CVD risk are great and aligned with the paper’s thesis that CVD should be managed comprehensively rather than by single risk factor. However, this is not mentioned in the introduction. I would suggest that the authors foreshadow these analyses by stating in the intro that in addition to the three groups (htn only, dm only, both), they also consider outcomes by total CVD risk.

2. Relatedly, I would suggest adding a sentence to the discussion on the main finding of the total CVD risk analyses.

3. The findings of appropriate management being lower among the combined group, especially when adding statins, is especially striking. One way to emphasize this would be to highlight in the introduction that appropriate management for those with multiple conditions is often more intensive than for those with a single condition. This would also increase the justification for focusing on the combined group.

4. In the discussion, the authors frame the problem of low pharmacological treatment as the result of access and cost. I would suggest the authors expand this to also consider the agency of individual patients - even when available and cheap, individuals may not initiate or adhere to treatment for a range of reasons (in fact I would guess that these are probably more important than access and cost...)

Reviewer #2: I have no further comments to the manuscript.

7. PLOS authors have the option to publish the peer review history of their article (what does this mean?). If published, this will include your full peer review and any attached files.

**Do you want your identity to be public for this peer review?** For information about this choice, including consent withdrawal, please see our Privacy Policy.

Reviewer #1: No

Reviewer #2: **Yes: **Shuduo Zhou
